# Volatile Organic Compounds in the *Azteca*/*Cecropia* Ant-Plant Symbiosis and the Role of Black Fungi

**DOI:** 10.3390/jof7100836

**Published:** 2021-10-06

**Authors:** Veronika E. Mayer, Sybren de Hoog, Simona M. Cristescu, Luciano Vera, Francesc X. Prenafeta-Boldú

**Affiliations:** 1Department of Botany and Biodiversity Research, Faculty of Life Sciences, University of Vienna, Rennweg 14, A-1030 Vienna, Austria; 2Center of Expertise in Mycology of Radboud University Medical Center/Canisius Wilhelmina Hospital, Geert Grooteplein 10 Zuid, 6525 GA Nijmegen, The Netherlands; sybren.dehoog@radboudumc.nl; 3Department of Analytical Chemistry and Chemometrics, Faculty of Science, Radboud University, Heyendaalseweg 135, 6525 AJ Nijmegen, The Netherlands; s.cristescu@science.ru.nl; 4Olfasense GmbH, Schauenburgerstr 116, 24118 Kiel, Germany; lvera@olfasense.com; 5Program of Sustainability in Biosystems, Institute of Agrifood Research and Technology (IRTA), Torre Marimon, E-08140 Caldes de Montbui, Spain

**Keywords:** ant-associated fungi, Chaetothyriales, volatile organic compounds, VOCs, *Cecropia-Azteca*, domatia, air biofilter hypothesis

## Abstract

Black fungi of the order Chaetothyriales are grown by many tropical plant-mutualistic ants as small so-called “patches” in their nests, which are located inside hollow structures provided by the host plant (“domatia”). These fungi are introduced and fostered by the ants, indicating that they are important for the colony. As several species of Chaetothyriales tolerate, adsorb, and metabolize toxic volatiles, we investigated the composition of volatile organic compounds (VOCs) of selected domatia in the *Azteca/Cecropia* ant-plant mutualism. Concentrations of VOCs in ant-inhabited domatia, empty domatia, and background air were compared. In total, 211 compounds belonging to 19 chemical families were identified. Ant-inhabited domatia were dominated by ketones with 2-heptanone, a well-known ant alarm semiochemical, as the most abundant volatile. Empty domatia were characterized by relatively high concentrations of the monoterpenes d-limonene, *p*-cymene and β-phellandrene, as well as the heterocyclic sulphur-containing compound, benzothiazole. These compounds have biocidal properties and are primarily biosynthesized by plants as a defense mechanism. Interestingly, most of the latter compounds were present at lower concentrations in ant inhabited domatia than in non-colonized ones. We suggest that Chaetothyriales may play a role in reducing the VOCs, underlining that the mutualistic nature of these fungi as VOCs accumulation might be detrimental for the ants, especially the larvae.

## 1. Introduction

Black fungi of the order Chaetothyriales have been hypothesized to constitute an important yet still poorly understood third partner in the trophic structure of ant/plant associations [1,2]. Chaetothyrialean fungi grow in dark brown to blackish patches in hollow plant structures (known as “domatia”) used by ants as their nesting sites (Figure 1E) or occur on ant-made cardboard such as “carton” used to make compartments in the hollow plant structures or to construct nests and tunnels [3,4,5]. It has been shown that foundress queens, before laying eggs, inoculate plant tissue scraped from new domatia walls with patch-material they brought from their mother colonies [6]. This patch-material consists of hyphal fragments and spores, mixed with organic matter, bacteria, and nematode eggs. In established colonies, worker ants were observed to nourish and groom patch material colonized by ant-associated Chaetothyriales [7,8]. At the molecular level, it was noted that domatia-colonizing taxa of Chaetothyriales, compared with their free-living counterparts, possess much smaller and more compact genomes, with considerable gene family contractions and losses [9]. They also had the highest content of repetitive elements detected in Chaetothyriales. A similar genomic pattern is known in ectomycorrhizal symbionts and has been referred to as the “symbiosis molecular toolbox” [10,11,12]. The interaction of ants with these fungal strains and their restriction to ant colonized habitats [1,2,13], suggest a mutualistic nature of ant-associated Chaetothyriales, rather than a fortuitous colonization of a suitable niche.

However, the role of these slow-growing, poorly competitive fungi [14] in arboreal ant nests is largely unknown. They may serve as food for growing larvae [15], as was suggested from isotope studies tracing labelled mycelia in feeding experiments. Alternatively, they may provide a certain protection for the nest against pathogenic microorganisms, since genome analyses indicate the production of antibiotics [9]. However, experiments to support the latter hypothesis are still lacking. In the present study we develop another, completely new hypothesis as to why ant-associated chaetothyrialean fungi may be actively introduced and fostered by arboreal ants.

In tropical ant-plant associations, ants live in domatia, the hollow plant structures which they use as nesting space. The domatia can be hollow stems, tubers, petioles, leaf pouches, or branches. The cavities are usually connected to outside air by only a single small entrance hole per domatium, which is just large enough for the ant workers to pass but hypothetically too small for an efficient air exchange (Figure 1A,B,D). Plants, as well as the ants, continually emit volatile organic compounds (VOCs) for many reasons, e.g., defense or communication [16,17,18]. Several of these volatiles are known for their toxic bioactivity, which is related to anti-herbivory or intruder deterrence [19,20]. Due to the reduced air exchange, inside the domatium VOCs may reach levels that are detrimental to eggs, larvae and pupae which still lack a protective exoskeleton. Consequently, the question arises of how domatia-inhabiting ants interact with such toxic substances. Studies have demonstrated that several species of Chaetothyriales tolerate or even assimilate certain toxic volatile compounds, such as alkylbenzenes, at relatively high concentrations [21], and have been used as biocatalysts in engineered air biofilters for the treatment of polluted air [22].

Based on the following three premises—(1) the need of plant-inhabiting ants to cope with plant-emitted toxic VOCs, (2) the deliberate cultivation of black fungi by plant-associated ants, and (3) the ability of some black fungi to metabolize VOCs—we hypothesize that Chaetothyriales in ant nests may be able to absorb and use plant- and ant-emitted volatiles as carbon and energy source. In view of data collection for this hypothesis, we sampled, analyzed, and quantified for the first time the VOCs emitted by plants and ants that accumulate inside the domatia of *Cecropia* plants inhabited by *Azteca* ants. Based on these results, we propose that ant-cultivated Chaetothyriales may be important for cleaning the domatia air from detrimental volatiles by acting as natural biofilters.

## 2. Materials and Methods

### 2.1. Sampling Campaign

Studied *Cecropia* trees (Urticaceae) came from the vicinity of the Biological Station La Gamba located near Golfito in Southwestern Costa Rica. The trees were identified as *C. peltata*, with the exception of a single *C. obtusifolia.* All of the trees were inhabited by *Azteca alfari* colonies (Formicidae, Dolichoderinae). Determination of the *Cecropia* species was according to [23], of the *Azteca* species according to [24,25]. In total, five *Cecropia* stems (2 m, 2.2 m, 3 m, 5.5 m, 6 m length) of five individuals were sampled. As two of the stems originated from the same plant, they were pooled. We collected the VOCs from a single domatium in the stem zone showing the highest ant activity. From two of the trees, we also sampled an uninhabited domatium near the apex (*n* = 2) to discriminate the plant-derived VOCs. Environmental air was sampled as control (*n* = 3). Further information on environmental parameters (time, °C, rh%) and domatium size, volume and position are given in Appendix A.

For the VOCs collection, we introduced the needle of a syringe into the entrance hole of the respective domatium and collected the headspace for 1 min with a membrane pump (G12/01, Thomas by GardnerDenver, Fürstenfeldbruck, Germany) using a flow of 0.1 L min^−1^ (Figure 1C,D). The air sample was then transferred into a stainless-steel tube (length: 3 in × 0.5 in. o.d.: 0.25 in. filled with a multisorbent bed of 350 mg of Tenax/Carbograph 5TD (Markes International Limited, Llantrisant, UK). This procedure was repeated 10 times with the same domatium in intervals of 30 min to allow VOC accumulation in between. In total, c. 1 L was collected. Controls consisted of 1 L of ambient air processed in the same way. Adsorption tubes had previously been conditioned by thermal cleaning (335 °C for 40 min) under a nitrogen flow rate of 50 mL min^−1^ (purity 99.9999%) by a tube conditioner device (TC20, Markes International Limited, Llantrisant, UK), and kept airtight until use.

### 2.2. Analytical Methods

Full quantitative scans on VOCs from the previously collected samples were carried out in a TD-GC-MS system. This instrument was composed of a thermal desorption unit (TD100-xr, Markes International, Llantrisant, UK), a gas chromatograph (TRACE 1310, Thermo Fisher Scientific), a mass spectrometer (ISQ 7000, Thermo Fisher Scientific). A mid polar TG-624 column was used for chromatographic separation (60 m, 0.25 mm, 1.5 µm; Thermo Fisher Scientific, Waltham, MA, USA). Specific parameters and experimental conditions of the analysis were previously published [26]. Desorption tubes were heated to 300 °C with a helium flow rate of 50 mL min^−1^ for 10 min (first desorption stage). Desorbed analytes were then directed to a hydrophobic general purpose cold trap (10 °C, thermoelectric cooling), filled with Tenax TA and graphitized carbon. After flash-heating of the cold trap to 320 °C during 5 min (second desorption stage), analytes were injected into the chromatographic column for further separation, which took an estimated 50 min. Molecules reaching the MS detector were fragmented by electron impact ionization (EI) at 70 eV at a mass range of 33–330 amu.

Deuterated toluene-d8 (Neochema GmbH, Bodenheim, Germany) was used as an external standard for quantification. This compound was injected (10 ng) into an independent thermal desorption tube and analyzed strictly following the same methodology as the samples. Given the sensitivity of the method, two unused thermal desorption tubes were analyzed as blanks, in order to exclude any potential contamination arising from these materials during the analysis. The deconvolution process for the chemical identification of the VOCs that were present in each analyzed sample was carried out with the software GC Analyzer (MsMetrix BV, Utrecht, The Netherlands). This algorithm automatically identified the compounds of the chromatogram, based on the NIST11 library [27]. Chemical identifications were confirmed with at least 80% certainly.

### 2.3. Statistical Data Analysis

Statistically significant differentiation in concentrations of specific chemical families of VOCs were evaluated by one-way analysis of variance (ANOVA). VOCs compositional profiles from the studied samples were analyzed by multivariate principal component analysis (PCA) by using Canoco Software (v5, Microcomputer Power, Ithaca, NY, USA). Only the identified compounds were considered, and their concentration values were log-transformed, standardized, and centered for their representation in a 2D biplot along with the sample scores.

## 3. Results and Discussion

### 3.1. Predominant Volatile Compounds in the Cecropia/Azteca Symbiosis

Previous studies revealed that Chaetothyriales are always found inside the hollow stem of *Cecropia* trees once inhabited with *Azteca* ants but never occur in stems without ant associates [3,6]. The chemical profile of VOCs and certain volatile inorganic compounds (VICs) was therefore characterized in air samples taken from the inside of *Cecropia* host-plant domatia, that were unoccupied (*n* = 3) and fully colonized by *Azteca alfari* (*n* = 3). Samples from the surrounding open air (*n* = 2), as well as from a scantly colonized domatium (*n* = 1), were taken as controls. When we opened the stem of the latter domatium after the air collection procedure, the queen was already dead, and the colony had therefore weakened. In total, 211 volatile compounds belonging to 19 chemical families were detected and identified (Appendix A). Of these compounds, 125 were identified in the open air of the sampled site at a total concentration ranging from 295 to 1038 µg m^−3^. The VOC emission of the fungi were not measured. In a previous study using black fungi, it was shown that volatile metabolites were only present at a concentration range <1 μg m^−3^ with a negligible impact on the total VOC composition [26].

In the atmospheric chemical background, the most abundant volatile compounds (average concentration higher than 5 µg m^−3^, Table 1) were the VICs sulphur dioxide (SO_2_) and carbon disulphide (CS_2_), as well as the VOCs 2-heptanone, acetic acid, 2-ethyl-1-hexanol, ethanol, acetaldehyde, diisobutyl phthalate, dibutyl phthalate, ethyl-cyclohexane, methyl acetate, and toluene. Sulphur-containing compounds are known to be naturally produced through volcanic activity, but may also biosynthesize in soils and sediments, and in plants. The average SO_2_ and CS_2_ concentration values measured in outdoor air in the present study were 366 and 38 µg m^−3^, which are well above the normal levels previously reported in rain forest of non-volcanic areas [28]. Consequently, a predominant geochemical origin is probable given the relative proximity of the studied site to the Turrialba active volcano [29]. As for the predominant VOCs of the background air, acetic acid and ethyl-cyclohexane have previously been identified as ubiquitous atmospheric compound in rain forests, derived primarily from biomass burning [30,31]. Acetic acid and other C2 volatile compounds (ethanol, acetone, and acetaldehyde) are also emitted by tree species in the rain forest, as a physiological response to anaerobic conditions in the roots [32], while the branched alcohol 2-ethyl-1-hexanol is an aroma compound in some flowering plants that positively correlates with flower-visiting insects [33]. In contrast, diisobutyl phthalate, dibutyl phthalate, and toluene are well known anthropogenic pollutants and their presence in a pristine environment is of concern. Phthalates are ubiquitous contaminants that are adsorbed onto the cuticles of insects, including ants from the rainforest, which have been proposed as good bioindicators for such type of pollution [34]. This bioaccumulation may explain the higher concentration of diisobutyl phthalate and dibutyl phthalate inside inhabited domatia seen in this study (Table 1). Toluene and other alkylbenzenes, on the other hand, are associated to fossil fuel pollution and have been found in pristine rain forests at increasing concentrations with proximity to urban areas [35].

The abundance and extremely high variability of the ketone 2-heptanone in the open air (1.7–176.5 µg m^−3^, Appendix A) might at least partially be attributed to the activity of *Azteca* ants. Inside ant-occupied domatia, the average concentration of this compound was about one to two orders of magnitude higher (5186 µg m^−3^ on average) than in the open air and represented the most abundant volatile found in this habitat. 2-Heptanone has historically been considered an alarm pheromone in ants, including those from the *Azteca* genus as reviewed by [36]. Some authors have also proposed that this chemical may act as an anesthetic on pests of social insects such as bees, enabling them to stun and eject intruders from the colony [37]. Inside inhabited domatia, ketones dominated. Of the 33 ketones identified in this study, 16 occurred exclusively and 5 at rather high concentrations in this niche, the majority being C5 to C8 methyl ketones (Appendix A). In addition to 2-heptanone as by far the most dominant volatile, 2-pentanone, 6-methyl-5-hepten-2-one, and 5-hepten-2-one also accumulated at average concentrations above 100 µg m^−3^. Previous studies on volatile emissions from *Azteca* ant nests confirmed the importance of 6-methyl-5-hepten-2-one as a defense pheromone [38]. 2-Heptanone is a common alarm pheromone secreted in the mandibular gland of many ant species [39,40,41,42]. When disturbed, *Azteca* workers release alarm pheromones also outside the nest while running along the *Cecropia* stem, which can be easily smelt by the human nose (Mayer, pers. observation). This is consistent with the fact that the odor threshold for 2-heptanone is as low as around 5 ppb [43], which is equivalent to 23 µg m^−3^ under normal conditions.

The concentrations of most ketones in the uninhabited domatia and background air were generally below 5 µg m^−3^, confirming that the compounds concerned were primarily produced by *Azteca* ants. Exceptions with higher concentrations were acetone, 2-pentanone and acetophenone in uninhabited domatia only, and 2-heptanone in background air and uninhabited domatia. Uninhabited domatia were, however, characterized by relatively high concentrations of the monoterpenes D-limonene, *p*-cymene and β-phellandrene (Table 1). Terpenic hydrocarbons are commonly present in the essential oils of a wide variety of plants, including domatia-forming species such as *Macaranga,* associated with *Crematogaster* from Southeast Asia [44], or *Piper* associated with *Pheidole* in Costa Rica [45]. Terpenes may generally have insecticidal properties [46,47,48,49], since they serve as feeding and oviposition deterrents to a large number of insects [50,51] as well as foraging cues for mammalian herbivores [52].

The heterocyclic compound benzothiazole was found inside empty domatia at concentrations (on average 40 µg m^−3^) that were significantly above that of the background air (on average 1.3 µg m^−3^). This molecule has been described as a xenobiotic and an environmental contaminant, but it also has a role as a plant secondary metabolite [53]. Benzothiazole exhibits antimicrobial effects against certain pathogenic fungi, nematodes, and insects [54], but it has also been proposed as a general attractant for garden ants in epiphytic associations, while it repels other ants of collecting seeds of ant garden plant associates [55,56]. It could thus well be that this compound is a specific cue for the recruitment of symbiotic *Azteca* ants to stimulate the colonization of vacant domatia. On the other hand, methyl 2-ethylpentanoate has been found in both inhabited and uninhabited domatia at concentrations that are significantly higher than the background air (Table 1). Methyl 2-ethylpentanoate is emitted by *Arabidopsis* when infected by aphids at high temperatures [57]. The high average concentration of diethyltoluamide in ambient air is due to one sample with an extraordinary concentration (Appendix A). Diethyltoluamide is a synthetic molecule that is the most common and widely used active ingredient in insect repellents [58], and the presence in ambient air samples might be the result of environmental pollution.

The total concentration values for ketones in the weakened *Azteca* colony under study were significantly lower than those found both in uninhabited domatia and in vital colonies (Table 1). 2-Heptanone was only slightly above the concentration of ambient air, some other ketones were even missing. This suggests that the colony members were too weak to produce and emit alarm pheromones such as 2-heptanone in large quantities. However, there were some specific compounds that were detected in that weak colony exclusively, or at relatively high concentrations, such as the aliphatic hydrocarbon octane (71.4 µg m^−3^), the sesquiterpene caryophyllene (9.4 µg m^−3^), and the ketones 1-cyclohexyl-ethanone (20.0 µg m^−3^) and 2-methyl-cyclopentanone (7.2 µg m^−3^) (Appendix A). While octane is known from sternal gland secretions of weaver ants [59], it is less clear whether caryophyllene is emitted from ants or plants. This compound is typically emitted after plant damage for interplant signaling. A study on the volatile organic compounds emitted by myrmecophytic and non-myrmecophytic species of the genus *Piper* revealed that caryophyllene was detected after stem damage, exclusively in the VOCs profile of myrmecophytic species [45]. Intense ant activity may damage the plant tissue of the domatia walls, which may lead to the release of caryophyllene. This may explain why caryophyllene was not detected in uninhabited domatia in our study. A survey on VOCs emitted by trees in a temperate Atlantic rainforest in Brazil reported the occurrence of different caryophyllene isomers in the related myrmecophytic species *Cecropia pachystachia* [60]. Whether this is also the case for *C. peltata* remains to be investigated.

Caryophyllene has also been proposed as a control agent of the south American ant species *Dorymyrmex thoracius* as an alternative of synthetic insecticides [61]. The authors showed that the compound was toxic to the insects at relatively high concentrations (LD_50_ = 1.49 µL L^−1^, for worker ants upon a 48-hour exposure). The genus *Dorymyrmex* belongs to the same subfamily Dolichoderinae as the *Azteca* ants in our study. As for the ketones, 2-methyl-cyclopentanone was identified long ago in the pygidial gland of *Azteca* spp. [62], while information on the natural occurrence of 1-cyclohexyl-ethanone is nearly absent. Recent studies have reported the latter compound as a minor component of the essential oils of certain plant species [63,64] and it was also found in the dung of wild ruminants [65]. Whether or not the specific occurrence of these compounds is coincidental or was causally related to the poor stage of this particular *Azteca* colony remains as an open question.

### 3.2. Multivariate Analysis of Chemical Profiles

A Principal Component Analysis (PCA) of the relative concentrations of the identified VOCs from our samples (Appendix A) was performed. Results are outlined in a 2D plot that represented the 61% of the total variance (Figure 2). In order to simplify the visualization of the PCA results, only volatile compounds that contributed to the ordination by at least a 70% fit on both axes have been displayed. Colonized domatia clustered along a gradient of increasing concentrations of ketones, which was primarily encompassed by the first PCA axis (42.7% of total variance). Besides ketones, methyl-cyclopentane contributed to the PCA ordination, which corroborates previous studies showing that cyclopentyl and methyl cyclopentenyl compounds are functional alert pheromones in *Azteca* oil secretions [38,62]. Despite repeated attempts, it was not possible to take air samples from colonized domatia without disturbing the ants, since the resident *Azteca* reacted immediately to manipulation of their habitat with distressed behavior. The observed emission pattern of ketones and cyclic compounds is therefore the product of the alarm-defensive system of the ants.

Uninhabited domatia, on the other hand, displayed quite a distinct chemical profile in that sample scores were mostly distributed along the second PCA axis (19.2% of total variance; Figure 2). This axis correlated with the concentration of several compounds that were relatively predominant in uninhabited domatia, such as thiophene and some alkylated derivatives from this compound. The heterocyclic sulphur-containing compounds occur in petroleum and are widely used as bulk chemicals, but they are also biosynthesized in nature. Biogenic thiophenes are characteristic secondary metabolites derived from plants acting as repellents, toxic substances, or by having anti-nutritional effects on herbivores [66]. Along with thiophenes, several secondary and branched alcohols and organic acids were also quite specific from uninhabited domatia (Figure 2). Several of these compounds have previously been found to be emitted by wide diversity of plant species and organs and might have both attractant and repellant functions to insects depending on the type of biological interaction [67,68,69].

Interestingly, the profile of volatiles in alarmed *Azteca* colonies negatively correlated with the concentration of alkylbenzenes, especially with that of o-xylene (Figure 2), with values that were generally lower than those present in uninhabited domatia and in background air (Appendix A). This trend also seems to occur with other chemical families (aldehydes and terpenes above all) and is discussed in the following section.

### 3.3. Role of Black Fungi in the Azteca/Cecropia Ant-Plant Symbiosis

Similarly to most ant-plant associations, domatia inside the living stems of *Cecropia* trees are connected to the environment by only small entrance holes (a single one per domatium), measuring 2–4 mm in diameter in the investigated trees. Such small openings are large enough for the *Azteca* worker ants to pass, but probably too small for appropriate air exchange. It would be easy for the strong mandibles of the ants to gnaw bigger holes and increase ventilation, if this would be favorable for the ant colony. However, apparently there are reasons to maintain small holes. Larger holes are more difficult to defend the colony against intruders and predators. Moreover, some ant species prefer hypercapnic (high carbon dioxide) conditions in their brood chambers, beyond atmospheric CO_2_ concentration levels [70]. This may also hold true for *Azteca* ants, as respiration inside the domatia is high due to the numerous ant individuals, the cohabitants in the colonies (scale insects, nematodes, fungi and bacteria), as well as metabolic processes in the patch (Mayer, pers. observ.).

A problem of poor ventilation might be the accumulation of plant volatiles that are continuously emitted from the tissue of the *Cecropia* host plant. Many of these chemicals, which have been detected in concentrations >10 µg³ in uninhabited domatia (Table 1), exhibit a broad toxicity range on several insect orders and have been tested for pest control in agriculture, protection of storage products and post-harvest protection. Aldehydes, e.g., acetaldehyde (13.6 µg³) and benzaldehyde (14.0 µg³ in uninhabited domatia) have been found to be toxic for aphids, mealybugs, thrips, and whitefly [71,72,73,74,75]. Carbon disulphide (145.6 µg³ in uninhabited domatia) has been used already since the 19th century as fumigant for insect control, especially against flour beetles [76,77]. Benzothiazole (41.4 µg³ in uninhabited domatia) is a promising insecticide against the red flour beetle for stored-product protection [78]. Terpenes are produced by plants for defense against herbivores [50,51] and many of them might be insecticidal. Cymene isomers (*p*-cymene was found with 28.9 µg³ in uninhabited domatia) have toxic effects on cigarette beetles, booklice [79], and red flour beetles [80]. D-Limonene (14.2 µg³ in uninhabited domatia) is an all-rounder against mealybugs, scale insects, whiteflies, aphids, rice weevil, corn rootworm, house fly, cat flees [46,49]. D-limonene is also used as pest control against the invasive fire ant *Solenopsis invicta* [81,82] and leaf cutter ants of the genus *Atta* [83]. Though the toxicity of these chemicals—with the exception of D-limonene—has not been tested against ants, they may at least be detrimental for the larvae and pupae as they still lack the protective exoskeleton and are confined continuously inside the domatium. The mechanism by which ants in domatia cope with the relatively high concentrations of the plant-emitted volatiles is still poorly understood.

Interestingly, the concentration of most of the aldehydes, aromatic compounds, sulphur-containing compounds and terpenes was on average higher in uninhabited domatia than in inhabited ones (Table 1andAppendix A). The average reduction in specific VOCs in inhabited domatia ranged between 8–97%, and was particularly relevant for some compounds known to have insecticidal properties, such as benzothiazole (97%), carbon disulphide (79%), acetaldehyde (70%), *p*-cymene (66%), D-limonene (63%) (Table 1). In summary, when subtracting the known insect alarm semiochemicals, the total reduction in the content of VOCs was of 54% (*p* = 0.026). It seems that inhabited domatia contain active mechanisms preventing accumulation of toxic VOCs and the hypothesis that black fungi play a role in this process is tempting. It was convincingly demonstrated that several species of Chaetothyriales are able to tolerate and even assimilate certain toxic volatile compounds, such as alkylbenzenes, at relatively high concentrations [21]. A competitive advantage of these fungi in extreme or toxic habitats has been suggested [84]. This trait has been investigated in the design of fungal bioreactors to treat industrial contaminants such as toluene, exploiting the resilience and extremophilic nature of chaetothyrialean fungi [22]. Recent studies have also demonstrated that the hydrophobic nature of fungal biomass composed by melanized species prompts the absorption of a wide range of volatile organic compounds, even if they are present at very low concentrations [26]. The behavior of the ants also indicates importance of the fungal patches. It has been shown that young *Azteca* foundress queens rapidly make a pile with plant tissue scratched from the domatia walls—before laying eggs—and inoculate it with patch material they had brought from their mother colonies [6]. In established colonies such patches where ant-associated Chaetothyriales grow are found in every single ant inhabited domatium, and worker ants were observed to nourish and groom them [7,8]. The role of Chaetothyriales for “air-cleaning” could explain why the ants are so eager to produce and maintain the patch colonized by these fungi.

In the *Azteca/Cecropia* ant-plant association *Azteca* ants make cardboard-like structures (“carton”) from the parenchyma layer of the domatia wall where they deposit the eggs, larvae, and pupae (Figure 1B and Figure 3). On the surface of this carton, numerous elongated, upright conidiophores in dense fascicles occur (Figure 3B). The assimilation of toxic VOCs by the conidiophores next to the brood may explain the high reduction in VOC concentrations in this ant-plant mutualism.

## 4. Conclusions

Our analysis of volatile organic compounds in the domatia inhabited by *Azteca* ants, and in those which had not yet been inhabited, shows that many aldehydes, aromatic compounds, sulphur-containing compounds and terpene had a considerably (77% in average) lower concentration in the headspace of inhabited domatia than in that of uninhabited ones. As the entrance holes to the domatia are rather small for an efficient ventilation, the reduction in concentrations rather points to an active mechanism mitigating the accumulation of detrimental volatile compounds. As several chaetothyrialean fungi are able to cope and even benefit from toxic volatiles, we suggest that these fungi might be responsible through biosorption and metabolic processes for the reduction in some VOCs inside the domatium and may therefore be actively nurtured by the ants. The accumulation of these chemicals could be deleterious to the vulnerable larval stage of the insects. Several related black fungi have already been tested in engineered biofilters for the treatment of VOCs polluted air. However, we are aware that the sample size in this study is still small, and the VOC profiles are still in-homogeneous. The fitness of the respective host plant, the ant colony size and individual behavior [85] as well as the developmental stage of the mycelium might influence the VOC composition of each sample. To further support the proposed “air-cleaning” hypothesis as a fundamental part of the symbiosis between plants, ants, and chaetothyrialean fungi, an increased sample size, and future experiments on the toxicity of VOCs for ant larvae, as well as on the usage of ant and plant emitted VOCs as C-source are needed.

## Figures and Tables

**Figure 1 jof-07-00836-f001:**
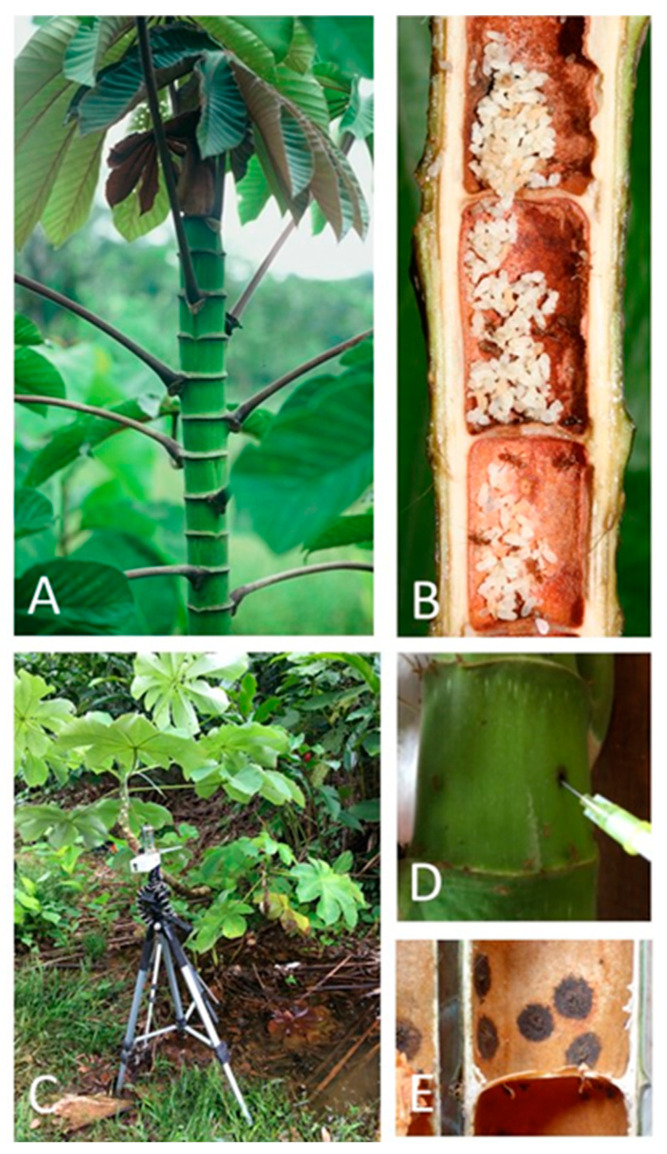
Habitus of *Cecropia obtusifolia* with the prominent visible internodes (**A**). The internodes are hollow and inhabited by *Azteca* ants (domatia) where they raise their larvae (**B**). Sampling device (**C**) and sampling of VOCs with a syringe needle through the entrance hole of the ant inhabitants (**D**). For sampling VOCs in the uninhabited domatia the hole was artificially made. (**E**) shows the patches where fungi are cultivated.

**Figure 2 jof-07-00836-f002:**
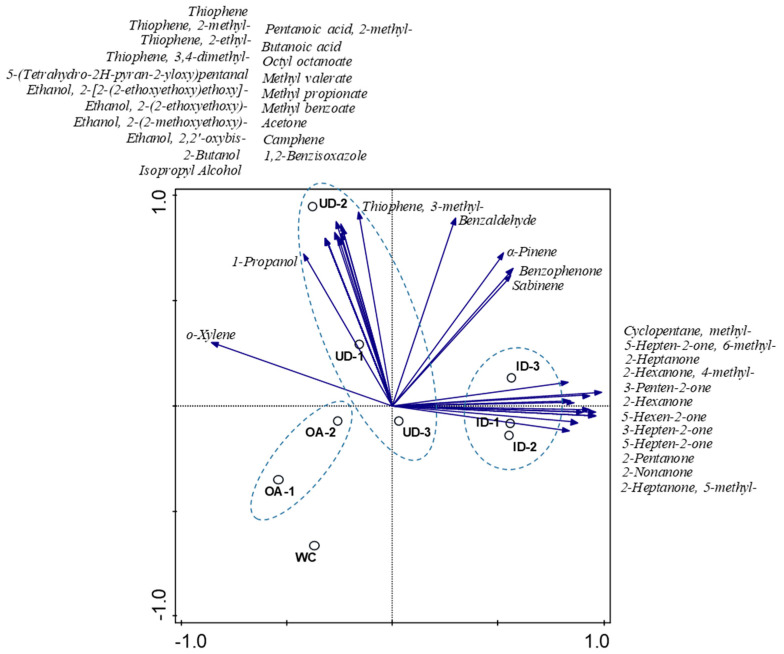
Principal Component Analysis (PCA) on the relative concentration of the identified volatile compounds (arrows) from the analyzed headspace samples from outdoor air (OA), uninhabited domatia (UD), inhabited domatia (ID), and a domatium inhabited with a weak colony (WC). The variance encompassed by the first and second PCA axis was 42.7% and 19.2%. Only volatile compounds that contributed to the ordination by at least a 70% fit on one of the axes have been displayed. Dotted lines mark the respective cluster.

**Figure 3 jof-07-00836-f003:**
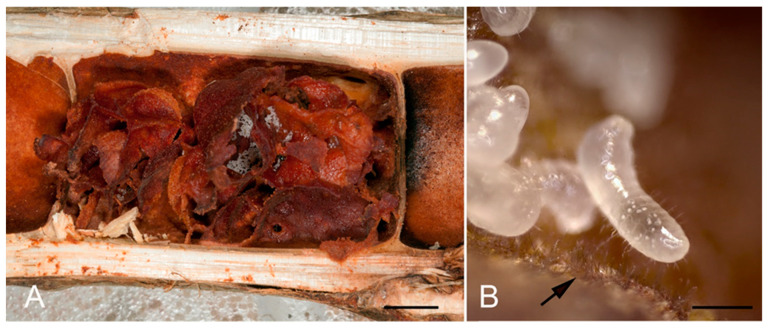
Domatium of *C. peltata* with cardboard-like structure (carton) with eggs young larvae of *A. alfari* (**A**). The carton surface with numerous elongated, upright conidiophores of chaetothyrialean fungi (**B**, arrow). Bars: 1 cm (**A**), 0.5 mm (**B**).

**Table 1 jof-07-00836-t001:** Total concentration of the volatile chemical families that are present in the outdoor air, and inside domatia of *Cecropia peltata* that were empty and colonized by *Azteca* ants. Specific chemicals present at concentrations above 5 µg m^−3^ in at least one of the samples have also been depicted. Chemical families are in bold.

Chemical Families and Compounds	Origin *^a^*	Outdoor Air *^b^*	Empty Domatia *^c^*		Colonized Domatia *^c^*		Weak Colony *^d^*	Reduction *^e^* (%)
**Alcohols**		**23.4**	**17.5**	**±6.8**	**13.4**	**±5.9**	**28.4**	**24**
Ethanol	P	8.4	3.5	±3.4	3.8	±1.6	0.8	43
2-Ethyl-1-hexanol	P	10.3	5.1	±2.5	9.0	±3.9	22.7	−76
**Aldehydes**		**30.5**	**51.7**	**±19.2**	**31.5**	**±10.9**	**19.9**	**39**
Acetaldehyde	P	8.3	11.7	±3.7	4.0	±4.9	4.5	66 *
Benzaldehyde	P	3.5	12.0	±9.9	8.8	±2.4	2.1	27
Nonanal	P	3.2	9.2	±10.3	4.5	±0.7	2.5	51
**Aliphatic Hydrocarbons**		**38.8**	**25.8**	**±15.8**	**39.3**	**±1.2**	**106.2**	**−52**
*n*-Hexane	P	1.8	0.8	±0.5	1.1	±0.3	7.3	−26
Octane	I	0.4	0.4	±0.1	0.4	±0.4	71.4	2
**Amines**		**0.1**	**0.1**	**±0.2**	**0.0**	**±0.0**	**0.0**	**100**
**Aromatic Alcohols**		**2.0**	**1.8**	**±1.8**	**1.0**	**±0.7**	**0.7**	**46**
**Aromatic Hydrocarbons**		**11.5**	**17.2**	**±13.8**	**7.6**	**±1.3**	**8.3**	**56**
Toluene	A	5.2	3.7	±2.4	3.1	±1.0	2.7	17
**Cyclic Hydrocarbons**		**7.9**	**7.3**	**±4.7**	**39.6**	**±29.0**	**3.6**	**−439**
Methyl-cyclopentane	I	0.0	1.0	±0.4	30.5	±28.7	0.0	−2808
Ethyl-cyclohexane	P	6.4	3.6	±3.8	6.3	±4.8	0.0	−75
**Esters**		**23.5**	**32.1**	**±23.3**	**83.0**	**±42.6**	**10.1**	**−158**
Methyl-acetate	P	5.6	1.7	±1.5	0.6	±0.6	0.4	64
Methyl−2-ethylpentanoate	P	0.0	0.1	±0.2	6.0	±3.9	2.2	−4798
Dibutyl-phthalate	A	7.1	16.5	±14.3	39.7	±20.9	0.0	−141
Diisobutyl phthalate	A	7.1	11.9	±10.4	31.8	±15.4	3.4	−167
**Ethers**		**1.4**	**1.8**	**±1.6**	**3.2**	**±1.8**	**2.9**	**−72**
**Furans**		**0.9**	**1.9**	**±2.8**	**2.7**	**±0.8**	**0.8**	**−38**
**Halogen-containing compounds**		**0.3**	**0.2**	**±0.0**	**0.1**	**±0.1**	**0.0**	**14**
**Heterogroups**		**0.6**	**15.7**	**±19.3**	**27.9**	**±28.1**	**2.6**	**−77**
Diethyltoluamide	A	0.6	14.8	±19.2	27.9	±28.1	0.0	89
**Ketones**		**73.8**	**406.0**	**±414.8**	**5853.5**	**±2022.6**	**172.9**	**−1342 ****
Acetone	P	3.3	14.0	±19.1	1.8	±3.2	0.0	87
2-Pentanone	I	1.3	10.5	±15.2	240.2	±196.9	2.7	−2179
2-Hexanone	I	0.2	1.3	±1.9	38.4	±22.5	0.5	−2791 **
2-Methyl-cyclopentanone	I	0.0	0.0		0.0		7.2	-
5-Hepten−2-one	I	0.0	1.1	±1.9	138.6	±65.5	0.0	−12,760 **
2-Heptanone	I	63.5	358.0	±412.2	5186.2	±1647.7	136.3	−1349 **
6-Methyl-5-hepten-2-one	I	2.0	6.3	±5.2	223.8	±79.0	0.4	−3429 **
1-Cyclohexyl-ethanone	I	0.0	0.0	±0.0	0.0	±0.0	20.0	-
Acetophenone	P	1.7	8.7	±12.4	2.0	±0.4	1.2	77
**Lactones**		**1.2**	**1.4**	**±0.3**	**0.8**	**±0.5**	**1.7**	**42**
**Nitrogen-containing compounds**		**5.6**	**7.4**	**±3.4**	**3.9**	**±1.3**	**8.1**	**47**
**Organic Acids**		**33.2**	**16.9**	**±13.2**	**11.9**	**±3.6**	**18.0**	**29**
Acetic acid	P	31.0	14.9	±12.1	9.8	±4.4	13.1	34
**Oxygen-containing compounds**		**0.3**	**2.1**	**±3.2**	**0.4**	**±0.1**	**0.7**	**79**
**Sulfur-containing compounds**		**408.2**	**672.1**	**±250.7**	**205.3**	**±300.5**	**59.3**	**69**
Sulfur dioxide	G	366.3	524.7	±266.1	171.7	±248.8	56.0	67
Carbon disulphide	G	38.4	113.6	±108.9	31.0	±49.4	1.8	73
Benzothiazole	P	1.5	27.9	±45.4	1.3	±0.3	1.6	95
**Terpenes**		**2.8**	**177.7**	**±163.5**	**41.1**	**±16.6**	**12.6**	**77**
α-Phellandrene	P	0.0	9.6	±16.2	1.6	±2.3	0.0	84
D-Limonene	P	1.7	86.5	±125.9	5.3	±2.7	1.0	94
*p*-Cymene	P	0.4	50.4	±45.9	9.8	±8.4	0.5	81
β-Phellandrene	P	0.0	9.0	±14.8	2.6	±2.6	0.0	72
Caryophyllene	P	0.0	0.0	±0.0	0.0	±0.0	9.4	-
Total concentration:		1151.1	1456.9	±435.3	6366.3	±2234.5	457.1	−337 **
Partial concentration *^f^*:		1021.2	1078.0	±114.0	501.1	±264.9	225.7	54 **

*^a^* Most likely origin of the volatile compounds: anthropogenic (A), geologic (G), insects (I), plants (P), *^b^* Average of two measurement points of ambient air. *^c^* Average and standard deviation of three different uninhabited and inhabited domatia with vital colonies. *^d^* Single measurement of a domatium inhabited with a weak colony. *^e^* Relative difference between uninhabited and inhabited domatia; such difference is significantly relevant at *p* < 0.05 (**) or 0.05 < *p* < 0.10 (*). *^f^* Excluding insect alarm semiochemicals. The chemical family is marked in bold.

## Data Availability

Not applicable.

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
