# Peer review of "Volatile Organic Compounds in the Azteca/Cecropia Ant-Plant Symbiosis and the Role of Black Fungi"

_jof, 2021, doi:10.3390/jof7100836_

Round 1

Reviewer 1 Report

The authors compared the volatiles in empty vs ant-inhabited domatia, finding that the latter included much more ketone semiochemicals while the former had more biocidal volatiles. They hypothesize that the Chaetothyriales fungus cultivated by the ants may contribute to scrubbing of the harmful VOCs in lieu of ventilation. The work is fascinating and introduces an interesting angle to understanding this ant-plant-fungal symbiotic relationship. The manuscript is well-written. I only have a few minor questions and suggestions that should be included in the manuscript:

Would the authors provide a method or citation for a method for identification of the targeted ant colonies?

Was there a sense of how much more fungus was present in healthy ant colonies compared to the weakened colony or compared to empty domatia?

Why was one sample of a weakened culture included in the statistical analysis of three empty domatia?

line 130- the degree sign is underlined

line 201- do the authors mean “in the colony” instead of “from the colony”?

Sometimes, there is a hyphen or underscore between “Table S” and the number.

Author Response

Response to reviewer 1

Would the authors provide a method or citation for a method for identification of the targeted ant colonies?

Sorry for not doing that earlier. I added the sources (see line 90 and 91). These are

  1. a) for the ants:

Longino, J.T. A taxonomic review of the genus Azteca (Hymenoptera: Formicidae) in Costa Rica and a global revision of the aurita group. Zootaxa 2007, 1491, 1-63, doi:https://doi.org/10.11646/zootaxa.1491.1.1,

Longino, J.T. Ants of Costa Rica. Available online: https://www.antwiki.org/wiki/Key_to_Costa_Rica_Azteca_queens,

  1. b) for the plants:

Berg, C.C.; Franco-Rosselli, P. Cecropia. In Flora Neotropica Monograph; New York Botanical Garden: 2005; Volume 94.

Was there a sense of how much more fungus was present in healthy ant colonies compared to the weakened colony or compared to empty domatia?

We know that in empty domatia chaetothyrealean fungi never occur. I added a sentence and respective references to clarify that (see line 151f “Previous studies revealed that Chaetothyriales are always found inside the hollow stem of Cecropia trees once inhabited with Azteca ants but never occur in stems without ant associates”).

However, we do not have information how much the fungal biomass differs between healthy colonies compared and weakened colonies. As the ants take care of the fungi we can only guess that reduced ant fitness has an impact on the fungi. 

Why was one sample of a weakened culture included in the statistical analysis of three empty domatia?

We found it very interesting that the VOC profile of the weakened colony was so different from the active ones.

line 130- the degree sign is underlined

This is now corrected.

line 201- do the authors mean “in the colony” instead of “from the colony”?

We mean indeed “from the colony” in the sense of “away from sth”.

Sometimes, there is a hyphen or underscore between “Table S” and the number.

This is now corrected.

Reviewer 2 Report

In this paper, an interesting new hypothesis has been tested with a sophisticated and elegant experimental design, i.e. . that black fungi associated with ants hep to detoxify the atmosphere of the ants inhabited domitia. The provided evidence seems to sustain this hypothesis.

Material and methods are described in depth. The Principal Component Analysis (PCA) of the identified VOCs shows a clear differential distribution of the 3 compartments.

Neithertheless, some questions still araise. These are the following ones:

What is the species of the black fungus associated with these ants?

Was it isolated and genetically described? At least morphologically described?

Please provide this information.

Was the black fungus present in empty domatia?

Is the VOCs emitted spectrum of the black fungus known?

Referring to table 1, it seems that the possibility that your black fungus emits VOCS has not been taken into account.

Are you sure black fungi do not emit VOCs at all?

What is the food resource used by the fungus inside the domitia?

Corrections

In the abstract, the expression “as the by far most dominant » must be replaced by “and by far the most dominant volatile”.

Lines 53-56. This sentence has to be reformulated for a better reading and understanding: “Together with the observed ant behavior and recurrent species restricted to these habitats (Voglmayr et al., 2011, Quan et al., 2020), this suggests a mutualistic nature of ant-associated Chaetothyriales, rather than a fortuitous colonization of a suitable niche.”

Citation in the text: Reference numbers should be placed in square brackets [ ], and placed before the punctuation; for example [1], [1–3] or [1,3].

Some back matter is missing: Author Contributions, Funding

The reference style does not respect the instructions for authors of the journal. This has to be corrected.

The references must be numbered in the order they appear in the text.

The font in the supplementary material must be Palatino Linotype as in the main manuscript

Author Response

Response to reviewer 2

What is the species of the black fungus associated with these ants?

There are still no species descriptions of the black fungus associated with Azteca. We have three pure cultures deposited at CBS and published the genotypes from environmental samples of >70 Azteca inhabited trees (Nepel, M.; Voglmayr, H.; Blatrix, R.; Longino, J.T.; Fiedler, K.; Schönenberger, J.; Mayer, V.E. Ant-cultivated Chaetothyriales in hollow stems of myrmecophytic Cecropia sp. trees – diversity and patterns. Fungal Ecology 2016, 23, 131-140, and Mayer, V.E.; Nepel, M.; Blatrix, R.; Oberhauser, F.B.; Fiedler, K.; Schönenberger, J.; Voglmayr, H. Transmission of fungal partners to incipient Cecropia-tree ant colonies. PLOS ONE 2018, 13, e0192207).

Was it isolated and genetically described? At least morphologically described? Please provide this information.

No, we did not isolate the fungi from the sampled trees. Isolation is very time consuming since the Chaetothyriales are very slow growing compared to the contaminants and during the VOC sampling we could not manage to concentrate on isolating the fungi. But we can refer to the genetic information already published (see above). In line 151f. I am now referring to these studies.

Was the black fungus present in empty domatia?

No. I added a sentence and references to explain this, see line 151f:

“Previous studies revealed that Chaetothyriales are always found inside the hollow stem of Cecropia trees once inhabited with Azteca ants but never occur in stems without ant associates [3,6].”

Is the VOCs emitted spectrum of the black fungus known?

No, we did not measure it and there is no chance to do it in the living system without applying an insecticide. We could only measure the VOC profile in the lab from pure cultures which is a completely different setting. But there is evidence that fungal emissions are very likely irrelevant because in a previous study by three of the co-authors it could be shown that the balance between the adsorption of environmental VOCs and emission of endogenous fungal VOCs is clearly in favor of adsorption. They used 3 phylogenetically distinct black fungi in these experiments, and the results were similar.

Prenafeta-Boldú FX, Roca N, Villatoro C, Vera L, and de Hoog GS. 2019. Prospective application of melanized fungi for the biofiltration of indoor air in closed bioregenerative systems. Journal of Hazardous Materials 361:1-9.

Referring to table 1, it seems that the possibility that your black fungus emits VOCS has not been taken into account.

See answer above

Are you sure black fungi do not emit VOCs at all?

I added a sentence referring to this question (see line 161f.):

“The VOC emission of the fungi were not measured. In a previous study it was shown that fungal metabolites were only present at a concentration range <1 μg m−3 with a negligible impact on the total VOC composition [26].”

What is the food resource used by the fungus inside the domitia?

This is not clear yet. Part of the food is probably the waste of the ants, but another part may be the volatiles. Our hypothesis, that also the ant-associated Chaetothyriales may be able to use some of the volatiles as carbon source. The ability to use aromatic hydrocarbons has been shown for other Chaetothyriales (Prenafeta-Boldú, F.X.; Summerbell, R.; Sybren de Hoog, G. Fungi growing on aromatic hydrocarbons: biotechnology's unexpected encounter with biohazard? FEMS Microbiology Reviews 2006, 30, 109-130, doi:10.1111/j.1574-6976.2005.00007.x.).

Corrections

In the abstract, the expression “as the by far most dominant » must be replaced by “and by far the most dominant volatile”.

The sentence has been reworded to:

Ant-inhabited domatia were dominated by ketones with 2-heptanone, a well-known ant alarm semiochemical, as the most abundant volatile. 

Lines 53-56. This sentence has to be reformulated for a better reading and understanding: “Together with the observed ant behavior and recurrent species restricted to these habitats (Voglmayr et al., 2011, Quan et al., 2020), this suggests a mutualistic nature of ant-associated Chaetothyriales, rather than a fortuitous colonization of a suitable niche.”

The sentence has been reworded to:

The interaction of ants with these fungal strains and their restriction to ant colonized habitats [1,2,13], suggest a mutualistic nature of ant-associated Chaetothyriales, rather than a fortuitous colonization of a suitable niche. (see line 51f.)

I hope that now it is better to understand what we mean.

Citation in the text: Reference numbers should be placed in square brackets [ ], and placed before the punctuation; for example [1], [1–3] or [1,3].

This is now corrected.

Some back matter is missing: Author Contributions, Funding

This is now added (see lines 414f.).

The reference style does not respect the instructions for authors of the journal. This has to be corrected.

This is now corrected.

The references must be numbered in the order they appear in the text.

This is now corrected.

The font in the supplementary material must be Palatino Linotype as in the main manuscript

This is now corrected.

Reviewer 3 Report

This paper introduces an intriguing hypothesis concerning the tripartite ecological association which is established by Azteca ants, chaetothyrialean fungi and Cecropia plants. It is very well written, and would undoubtedly rise the attention of readers. However, I have perplexities concerning the statistical analysis based on data reported in Table 1. In fact, the standard deviation for many values is quite high, which affects significance considering that means resulted from 3 measurements only. Sometimes, SD is even higher than the mean itself (e.g. nonanal 9.2±10.3; acetaldehyde 4.0±4.9). Quite clearly, this derives from the wide variation characterizing the samples, which are obviously non-homogeneous. In fact, the atmosphere composition in domatia is likey to change due to several factors which cannot be taken into account in the statistical analysis to minimize error. As an example, besides disturbance of ants during sampling, at some extent the VOC assortment could be influenced by the developmental stage of the mycelium, which is not necessarily uniform in all domatia. Moreover, the associated fungus might directly contribute to this assortment in a way that is not necessarily homogeneous. Actually, authors themselves recognize (lines 250-251) that means are possibly influenced by exaggerate values in some samples, which unavoidably impairs significance. I am inclined to think that authors decided not to increase the number of samples, which would have somehow reduced this error, by technical problems associated with processing. Whatever the reason, the statistical analysis they performed appears to be biased, and makes it impossible to reach coherent conclusions.

Author Response

Response to reviewer 3

However, I have perplexities concerning the statistical analysis based on data reported in Table 1. In fact, the standard deviation for many values is quite high, which affects significance considering that means resulted from 3 measurements only. Sometimes, SD is even higher than the mean itself (e.g. nonanal 9.2±10.3; acetaldehyde 4.0±4.9). Quite clearly, this derives from the wide variation characterizing the samples, which are obviously non-homogeneous.

Yes, your concern is justified.

We are in fact aware of the small sample size and the high variation in the VOC profiles. It was in fact not so clever to use in Table 1 SD instead of SE. However, the variability is anyway obvious, and we emphasized from the very beginning that we “hypothesize” and “propose” (see line 53f: “Based on these results, we propose that ant-cultivated Chaetothyriales may be important for cleaning the domatia air from detrimental volatiles by acting as natural biofilters.”).

The problem with ant-plant-fungus systems in general and the Cecropia-Azteca system in particular, is, that they do not only comprise these three players but up to 8 different organismic groups. In the Azteca-Cecropia system the players are plants, ants, fungi, bacteria, nematodes, dipteran larvae, mites, and pseudo-coccids, all of them have different life histories and are somehow interacting. At the moment we argue predominantly with trends and our profound knowledge of ant-plant systems and the ecology and evolution of black yeasts.

In fact, the atmosphere composition in domatia is likey to change due to several factors which cannot be taken into account in the statistical analysis to minimize error. As an example, besides disturbance of ants during sampling, at some extent the VOC assortment could be influenced by the developmental stage of the mycelium, which is not necessarily uniform in all domatia.

This is also a very good point.

The developmental stage of the mycelium is not at all uniform, not in all domatia of the same host plant and not in the individual colonies. The activity of the ant colony seems to be an important factor, the humidity inside the stem. Even if two trees look the same from outside, after opening the stem the fungal patches and the carton may look very different inside.

Moreover, the associated fungus might directly contribute to this assortment in a way that is not necessarily homogeneous.

In a previous study (Prenafeta-Boldú, F.X.; Roca, N.; Villatoro, C.; Vera, L.; de Hoog, G.S. Prospective application of melanized fungi for the biofiltration of indoor air in closed bioregenerative systems. Journal of Hazardous Materials 2019, 361, 1-9) it was shown that black fungi emit only few VOCs and only in low concentrations (<1 μg m−3). Further, fungi, and particularly the black yeasts, have a quite hydrophobic biomass and adsorb therefore, a wide range of VOCs. The balance between the adsorption of environmental VOCs and emission of endogenous ones is clearly in favor of adsorption of environmental VOCs. Based on this results we think that fungal emissions are very likely irrelevant in our study.

I am inclined to think that authors decided not to increase the number of samples, which would have somehow reduced this error, by technical problems associated with processing. Whatever the reason, the statistical analysis they performed appears to be biased, and makes it impossible to reach coherent conclusions.

The sampling in the field was indeed difficult and a major issue was that we could not test the method at home before leaving to the tropics. Such a system does simply not exist in Europe. We were fortunate that we could get what we show in the manuscript. You are right that we need much more samples to get coherent results and we need additional experiments to strengthen our hypothesis. I added a sentence in the conclusion to point this out (see line 403f.):

“However, we are aware that the sample size in this study is still small and the VOC profiles are still in-homogeneous. The fitness of the respective host plant, the ant colony size and individual behavior [86] as well as the developmental stage of the mycelium might influence the VOC composition of each sample.”

I hope that we sufficiently addressed your concerns. Thank you very much for digging deeper into the manuscript. It helped to once more discuss and clarify our research questions and next steps.

Round 2

Reviewer 3 Report

After reading authors' response, I can only say that it is a bit awkward that they recognize flaws in their paper and still think it can be published like it is. Statistical analysis is not a compulsory tool to be used in describing a biological system. But, if it has to be used, it must be used correctly in order to provide meaningful results. And it is not the case of this paper.